# Application of machine learning techniques for regional bias correction of SWE estimates in Ontario, Canada

Fraser King[1], Andre R. Erler[2], Steven K. Frey[2,3], and Christopher G. Fletcher[1]

[1]Dept. of Geography & Environmental Management, University of Waterloo, Ontario, Canada
[2]Aquanty, Waterloo, Ontario, Canada
[3]Dept. of Earth & Environmental Sciences, University of Waterloo, Ontario, Canada

**Correspondence:** Fraser King (fdmking@uwaterloo.ca)

**Abstract.** Snow is a critical contributor to Ontario's water-energy budget with impacts on water resource management and flood forecasting. Snow water equivalent (SWE) describes the amount of water stored in a snowpack and is important in deriving estimates of snowmelt. However, only a limited number of sparsely distributed snow survey sites (n=383) exist throughout Ontario. The SNOw Data Assimilation System (SNODAS) is a daily, 1 km gridded SWE product that provides uniform spatial coverage across this region; however, we show here that SWE estimates from SNODAS display a strong positive mean bias of 50% (16 mm SWE) when compared to in situ observations from 2011 to 2018. This study evaluates multiple statistical techniques of varying complexity, including simple subtraction, linear regression and machine learning methods to bias-correct SNODAS SWE estimates using absolute mean bias and RMSE as evaluation criteria. Results show that the random forest (RF) algorithm is most effective at reducing bias in SNODAS SWE, with an absolute mean bias of 0.2 mm and RMSE of 3.64 mm when compared with in situ observations. Other methods, such as mean bias subtraction and linear regression, are somewhat effective at bias reduction however, only the RF method captures the nonlinearity in the bias, and its interannual variability. Applying the RF model to the full spatio-temporal domain shows that the SWE bias is largest before 2015, during the spring melt period, north of 44.5° N and East (downwind) of the Great Lakes. As an independent validation, we also compare estimated snowmelt volumes with observed hydrographs, and demonstrate that uncorrected SNODAS SWE is associated with unrealistically large volumes at the time of the spring freshet, while bias-corrected SWE values are highly consistent with observed discharge volumes.

## 1 Introduction

Snow melt is an important factor for determining flood risk in many regions within both Northern latitudes and higher elevation across Europe and North America (Berghuijs et al., 2019, 2016; Buttle et al., 2016). Accordingly, predicting the impact of snowmelt on flooding is contingent on having reasonable spatially distributed estimates of the snowpack snow water equivalent (SWE). SWE is the amount of liquid water that is produced from completely and instantly melting a snowpack and is defined in terms of snowpack depth and bulk density (or, equivalently, mass per unit area). Traditionally, ground-based observations have been used to assess and quantify SWE; however, such an approach does not always capture the full spatial variability.

In Canada, large scale snowmelt is often a key driver of flooding across much of the southern, and more populated, parts of the country (Buttle et al., 2016), and one can posit that an improved ability to characterize snowpack SWE would allow better characterization of flood risk, propagation, and duration. Particularly, within the Canadian provinces of Ontario and Quebec, snowmelt and rain-on-snow events are the most frequent initiators of flooding (Buttle et al., 2016; Irvine and Drake, 1987).

Regional flood danger was realized in a 2017 flood across Southern Quebec which damaged over four thousand homes and lead to approximately 200 million dollars worth of insured damages (Davies, 2017). Additional serious snowmelt and rain-on-snow induced flooding has occurred in Southern Quebec and Ontario as recently as spring 2019 (Floodlist, 2019). These recent events indicate that even though future SWE is projected to decline in south-central Canada on account of warmer winters, snowmelt will be a major flood factor in this region for the foreseeable future, with a strong likelihood of an increase in the

frequency of rain-on-snow events (Byun et al., 2019).

Simulation and operational forecasting of flood risk necessitates insight on the contribution of snowmelt to the active component of the terrestrial hydrologic cycle. This is particularly important if snowmelt is anticipated to influence flood behavior (Li and Simonovic, 2002; Bokhorst et al., 2016), and the modelling tools employed for such applications include the capability to simulate snowpack dynamics (Anderson, 1976; Jordan, 1991). However, due to the high spatial and temporal variability of

the snowpack, combined with the sparse distribution of in situ observations, it is difficult to properly initialize and validate forecast models. For this reason, data assimilation products such as SNODAS, which operates at 1 km spatial resolution (Carroll et al., 2001) with a daily update cycle, offer an attractive option for characterizing SWE. For example, Shen and Anagnostou (2017) employed SNODAS data aggregated to 8 km x 8 km grid cells as a validation dataset for their hydrologic model of the $20,000\ km^2$ Connecticut River Basin, wherein SWE was simulated using an energy balance approach. In CONUS scale work,

Vuyovich et al. (2014) utilized SNODAS SWE as a comparative benchmark in their evaluation of SWE derived from passive microwave satellite sensors, wherein the data were aggregated at the scale of watersheds with an average size of $3,700\ km^2$.

An inherent challenge with using SNODAS as either a validation target or as direct forcing data for hydrologic modelling is that SNODAS itself may contain biases or errors that will in turn propagate through to the model outputs. In this context, the motivation for this work derives from an initial comparison between SNODAS and an independent set of in situ SWE surveys

throughout Ontario (section 3.1), which suggested a positive bias of approximately 50% in the SNODAS estimates. Wrzesien et al. (2017) performed a comparison between SNODAS and in situ SWE over alpine regions in North America and found SNODAS performed best in areas with a high density of in situ measurements, however SNODAS still exhibited a general overestimation of SWE. Additional recent studies by Leach et al. (2018), Lv et al. (2019) and Zahmatkesh et al. (2019) also suggest similar positive biases in SNODAS SWE estimates throughout other North American regions. This work builds on the

comparison methods outlined in previous bias correction studies by Li et al. (2010), Themeßl et al. (2011) and Teutschbein and Seibert (2012), to examine an ensemble of bias correction techniques, quantify the skill of each model, and apply the model over a larger spatio-temporal domain to produce a gridded bias corrected SWE product. Biases in initial SWE estimates constitute a major source of uncertainty in hydrologic modelling (Islam and Déry, 2017); yet, at present this important influence of SNODAS biases on simulated hydrologic behavior and flood magnitude is not well understood. Accordingly, the primary

objectives of this work are to evaluate:

1. Biases in SNODAS across flood prone regions of Ontario, Canada.

2. The effectiveness of SNODAS bias correction from simple subtraction methods to more sophisticated machine learning techniques.

3. The relationship between the regional water balance and snowmelt estimates from SNODAS SWE and bias-corrected SWE.

Section 3.1 quantifies current biases between SNODAS and in situ SWE estimates throughout Ontario. Sections 3.2 and 3.3 present evaluations of several statistical methods for bias correction, to determine whether machine learning techniques offer improved performance compared to more traditional linear methods. In Section 4, the best-performing bias correction model is then applied across the full spatio-temporal domain to create a daily, bias corrected SWE dataset which can be compared with the uncorrected SNODAS record. Differences between these two products can provide insights into where and when the bias is strongest. Finally, in Section 4.1 the impact of these difference on snowmelt volume and the influence on the regional water balance is evaluated in three representative catchment areas.

## 2 Data and methods

### 2.1 In situ data

In situ snow survey data is retrieved from a dataset created by the Climate Research Division of Environment and Climate Change Canada (ECCC), which has been updated to include observations up to the end of 2017 (ECCC, 2000). This dataset includes snow survey measurements from approximately 33 Conservation Authorities (CAs) at 383 locations throughout Ontario between $41°$ N and $49.5°$ N and $-87.875°$ E and $-73.375°$ E. The locations of these survey sites are marked in Fig. 1 a along with an outline of our Ontario study region. Survey site locations are selected based on recommendations set forth by the Conservation Authorities and Water Management Branch which suggest that measurement sites should be selected based on their representativeness of the surrounding region, in easily accessible locations free from the effects of excess wind drifting (Authorities, 1985). The snow surveys provide an estimate of SWE calculated as an average of 10 individual snow core SWE samples taken over 10 meters at each survey site (Authorities, 1985). These observations are recorded bi-weekly around the $1^{st}$ and $15^{th}$ of each month from November to May and have been recorded since 1933, but for this study only data from January 2011 until December 2017 have been considered. Snow survey density is higher in the southernmost portion of Ontario (below $44.5°$ N) with 189 survey locations closely grouped near the United States (US) border between Lake Huron and Lake Ontario. A similar survey count ($n = 194$) exists above $44.5°$ N but with sparser spatial coverage due to the region's larger size and lower population density.

The measurement tools used for retrieving in situ observations vary between locations over the time span of our study (Authorities, 1985; Sturm et al., 2010). Common strategies for collecting SWE measurements by hand include the use of snow corers which are portable, handheld tubes that are inserted into the snowpack down to the soil layer and weighed to retrieve a

SWE estimate at that point within the snowpack (López-Moreno et al., 2013). An Ontario snow inventory summary completed by Metcalfe (2018) provided a questionnaire to 265 of the snow survey sites (with a 67% response rate) and found that the Federal Snow Sampler (also known as the Mount Rose sampler) was used at 94% of sites and the ESC-30 was used at the remaining 6% of locations. Methods of SWE measurement also varied with 62% using a calibrated spring balance in the field, 30% using a digital balance in the field, 6% grouping snow samples into a container and weighing in the field, and 2% bagging the samples and weighing them later (Metcalfe, 2018). Although an important observational metric, in situ measurements of SWE take, on average, 20 times longer than snow depth measurements, and due to the additional time investment this often results in poor spatial and temporal data coverage of SWE measurements across large regions (Sturm et al., 2010).

## 2.2 Gridded SWE products

### 2.2.1 The Snow Data Assimilation System (SNODAS)

SNODAS is a gridded modeling and data assimilation dataset produced by the National Oceanic and Atmospheric Administration (NOAA) National Weather Service's Operational Hydrologic Remote Sensing Center (NOHRSC) (Barrett, 2003). SNODAS provides a physically consistent framework for assimilating snow data from nearly all available North American airborne, satellite and ground station sources with a Numerical Weather Prediction (NWP) snow model (Dawson et al., 2016). Produced at 1 km resolution, SNODAS covers the continental US from approximately $25.95°$ N to $52.87°$ N and overlaps with portions of Canada including our study region (Azar et al., 2008). Daily estimates are provided from September 2003 to January 2018, however Ontario was only included within the assimilation domain starting in January 2011, providing seven years of overlapping data with the in situ SWE measurements. Additional SNODAS product details are described in Table 1.

SNODAS is composed of three primary components: the data ingestion pipeline which handles data quality control and downscaling from the NWP model forecasts, the snow mass and energy-balance model which calculates hourly snowpack property estimates, and the data assimilation scheme which updates the model state with observational snow data (Carroll et al., 2001). In order to prescribe forcing data for the snow model, SNODAS makes use of the Rapid Refresh (RAP) and High-resolution Rapid Refresh (HRRR) NWP systems, deployed by the National Centers for Environmental Prediction (NCEP) to produce high accuracy, hourly numerical weather forecasts (Benjamin et al., 2016). RAP/HRRR produces analyses and short-term forecasts of precipitation, pressure, temperature, wind and relative humidity which are corrected using station and radar data, downscaled, assessed for quality and then used to force the SNODAS snow model (Barrett, 2003). SNODAS uses a spatially distributed multi-layer mass and energy-balance snow model with 3 snow layers and 2 soil layers (Carroll et al., 2001). The snow model calculates snowpack SWE, temperature, thickness and liquid water fraction within each snow layer and produces an estimate of total SWE, runoff melt (from the base of the snowpack), as well as estimates of exchange fluxes with the atmosphere. Thermal properties of the snowpack are simulated using similar techniques to SNTHERM89 as described in Jordan (1991). After applying the surface and atmospheric forecasts from RAP/HRRR, the snow model is run at an hourly timestep, with mass and energy balance calculated at each grid cell (Barrett, 2003).

A simple nudging method (Newtonian Relaxation Procedure) is then used to update model SWE estimates with assimilated ground-based, airborne and satellite snow observations (Boniface et al., 2015). This technique examines differences between numerical model SWE estimates and assimilated observations to identify regions with significant differences (Clow and Nanus, 2011). Although many existing snow cover and SWE datasets are assimilated by SNODAS, we note that the in situ snow survey dataset employed in this study is not assimilated by SNODAS. Differences between the model estimates and observations are then interpolated to produce nudging fields (an increment used to *nudge* model estimates closer to observations) and the model is re-run for the previous 6 hours. Each hourly increment during this period is nudged using the previously computed nudging fields to produce the final SWE estimate for each grid cell, updated using assimilated observational datasets (Barrett, 2003). Previous studies by Frankenstein et al. (2008) and Rutter et al. (2008) have suggested that SNODAS strongly benefits from this data assimilation step with densely observed locations displaying high quality SWE estimates in SNODAS when compared with in situ measurements.

### 2.2.2 NRCan ANUSPLIN data

During the development of the bias-correction methods, a gridded, monthly climatology (spanning 1981-2010) of 2 meter air temperature and total precipitation was employed. This dataset was developed by the Canadian Forestry Service (CFS), which is a division of Natural Resources Canada (NRCan); it will be henceforth referred to as the NRCan dataset. The NRCan dataset is generated through the use of thin-plate (Laplacian) smoothing splines which interpolates point observations over a grid as implemented in the ANUSPLIN (Australian National University SPLINe) climate modeling package (Hutchinson et al., 1991; McKenney et al., 2011). The NRCan product provides additional gridded estimates of snowpack height, 2 meter air temperature and total precipitation throughout Ontario (Table 1). This product has a spatial resolution of approximately 10 km and provides monthly normal estimates of surface parameters from January 1981 to December 2010. This observational time frame overlaps with in situ survey measurements, however NRCan data ends (December 2010) just before SNODAS becomes available in this region (January 2011) which is an additional source of uncertainty (see section 4.2). The datasets used in the generation of this product are independent from both the SNODAS and the snow survey datasets.

### 2.3 Statistical methods for bias correction

A set of statistical methods that have previously been applied to bias correction in different contexts are analysed in this study to identify the method which displays the highest performance in reducing the bias between SNODAS SWE and in situ observations over our study period. The methods examined include: mean bias subtraction (MBS), simple linear regression (SLR), decision trees (DT) and random forest (RF). All models (excluding MBS) are implemented using the scikit-learn Python package which includes built in linear regression and machine learning modules (Pedregosa et al., 2011). For MBS, the average difference in SWE between SNODAS and in situ is calculated and then subtracted from each SNODAS estimate to produce a bias corrected dataset. More formally, mean bias (MB) is defined as: $MB = \frac{1}{n} \sum_{i=1}^{n} (x_i - z_i)$ where $x_i$ and $z_i$ are the respective daily SNODAS and in situ SWE measurements, and $n$ is the number of measurements over the study period. The linear regression techniques used in this study conform to the least squares general regression model which relates a response

variable $y$ to a linear combination of $n$ explanatory $x$-variable predictors $\hat{y} = b_0 + b_1 x_1 + b_2 x_2 + ... + b_n x_n$ where $b_0$ is the model intercept and $b_1$ to $b_n$ are predictor coefficients. Both, simple linear regression (SLR) using a single explanatory variable, and multiple linear regression (MLR) with numerous explanatory variables are considered in this study. A single SLR model is applied to all stations in our study region using daily SWE from SNODAS as the sole predictor and in situ snow survey SWE estimates as the response variable. MLR, DT and RF methods all use the full list of predictors outlined in Table 2 to predict in situ SWE.

A decision tree is a flowchart-like data structure, wherein the decision making process begins at the root node (the top of the tree) followed by a series of cascading decisions based on the included model predictors until the terminal leaves are reached at the bottom of the tree which represent the regression estimate of the response variable. As implied by its name, the random forest regression model is an ensemble of decision trees that are generated during model training (Azar et al., 2008). Each tree included in the RF model ensemble is generated from a randomized subset of the available training data, coupled with a randomized subset of the predictor variables (Breiman, 2001; Grömping, 2009). This inherent randomness improves the learning process of this technique, but also contributes to uncertainty in the accuracy of an individual tree (Barnett et al., 1988). The ensemble approach used in RF accounts for and minimizes the uncertainty present in an individual decision tree by calculating the mean prediction from all trees in the ensemble. Our RF model is run with a forest size of 100 trees in its ensemble, both RF and DT methods use a maximum tree depth of 15 (the maximum number of decisions before determining an estimate for the response variable) and each tree was allowed to grow to its full extent, with no set number of maximal terminal nodes (no set maximum number of leaf nodes). These model parameters were obtained through a brute-force grid search hyperparamaterization of the RF model, which nudged each parameter value and examined changes in model accuracy and impacts on computational efficiency to select optimal values for running the model. A variety of regression predictors were considered for use in this study, including land use parameters, elevation, and indicators of general climate. The final set of predictors for all methods (shown in Table 2) was selected based on the (non-zero) model importance score for each variable in the RF model summary output.

To mitigate against model overfitting, the data for the RF and DT models are randomly split into a training set composed of 75% of the values, and a testing set which comprises the remaining 25%. Additionally, a separate 10-fold cross validation (CV) resampling procedure was applied to further evaluate model performance on unseen data. The CV K-fold splits the full dataset in time into 10 consecutive groups of samples which are held constant for the full CV procedure. We then train the model on each combination of $k - 1$ folds and their performance is calculated as the average of all training and testing scores for each K-fold split. The fold value of $k = 10$ was selected as a compromise between the size of the training sample and the computational overhead, and is a typical choice for similar applications. (James et al., 2013).

A range of metrics have been considered in order to determine whether additional performance can be gained from using more sophisticated statistical methods over traditional approaches. To assess model skill, we have selected absolute mean bias (accuracy) and RMSE (precision) as our model performance criteria, as these properties have demonstrated effectiveness in previous studies for assessing the capabilities of competing bias correction methods in the geosciences (Cannon et al., 2015; Grossi et al., 2017; Li et al., 2010).

In addition to applying each model to the full set of in situ survey sites for the full time span, we also run spatially and temporally partitioned models to assess changes in performance over specific regions and periods. Partitioning is applied spatially by separating the set of in situ measurement locations into northern and southern regions at 44.5° N to help account for snow survey density differences between the two regions of Ontario as described in section 2.1. Model performance is also analysed temporally with training restricted to different portions of the snow season: December to February (DJF), March to May (MAM) and the combined period: December to May (DJFMAM). All models and partitioned datasets were trained on 75% of the data and tested on the remaining 25% (excluding MBS which does not include a model training step).

## 3 Bias correction results

### 3.1 Quantifying biases in SNODAS SWE

Initial comparisons between current SNODAS SWE estimates and in situ observations throughout Ontario describe, on average, a positive absolute mean bias of 50% in the SNODAS estimates from 2011 to 2017. Additionally, the snow survey sites in Fig. 1 a display a pattern of strong relative mean bias present in the SNODAS estimates at the majority of survey locations. Relative mean bias (RMB) is defined as $RMB = \frac{1}{n} \sum_{i=1}^{n} \frac{x_i - z_i}{z_i} * 100$ where $x_i$ and $z_i$ are the respective daily SNODAS and in situ SWE measurements, and $n$ is the observation count. This relative bias is positive at 212 of the 383 measurement sites and rises above +100% relative bias at 67 locations. These sites with a strong relative bias also generally exhibit a strongly positive absolute mean bias, with SNODAS overestimating SWE by over 100 mm SWE at many of these locations. The sites with the strongest relative and absolute mean biases are typically grouped together in the northern portion of the study region above 44.5° N, as well as in areas East of both Lake Huron and Lake Superior.

There also exists a strong temporal bias in the bi-weekly SWE estimates from SNODAS when compared with in situ (Fig. 1 b). This bias is strongest during the first half of our study period until the beginning of 2015 where, although SNODAS estimates are generally still higher on average (by approximately 5 mm SWE), the overall absolute mean bias is reduced. If we consider the full temporal domain, the absolute mean bias in the SNODAS estimates is approximately 16 mm SWE, which corresponds to a 50% increase compared to that of the in situ SWE observations. The change in bias between the first and second half of our study period implies a change in the data assimilation system used by SNODAS, wherein new datasets are assimilated into the system to further reduce model error.

### 3.2 Simple subtraction and regression techniques

In the following section, the performance of four bias correction techniques will be discussed. The progression of mean bias and RMSE over the two study regions and three time periods is summarized in Fig 2, timeseries summary metrics for the full region are shown in Fig. 3, and the spatial pattern of (remaining) absolute mean biases at snow survey sites is shown in Fig. 4; the timeseries of corrected and uncorrected, domain-averaged SWE (along with the 95% confidence intervals based on each sample) is shown in Fig. 5 for our full study period.

### 3.2.1 Mean bias subtraction

We begin by quantifying how well SNODAS SWE biases can be reduced through a simple subtraction of its mean bias. Since this method is constructed through the removal of the mean bias in the SNODAS data record, MBS reduces the absolute mean bias between SNODAS and in situ to zero when averaged over all regions across all seasons as shown in Fig 2. Although zero absolute mean bias gives the appearance of strong performance, residual biases still remain at individual days and months of the MBS corrected dataset.

The RMSE of the resulting bias corrected dataset is only slightly reduced compared to the default RMSE between SNODAS and in situ SWE. The largest decreases in RMSE of approximately 15 mm SWE (30%) occur in the northern region, with a more muted reduction throughout the southern region (approximately 2 mm SWE (10%) on average). Similar reductions in RMSE follow when this technique is applied over all years for both regions as shown in Fig. 3. RMSE is reduced from the default SNODAS value of 27.45 mm SWE to 21.9 mm SWE, which is an improvement of approximately 20%.

As was previously noted, this technique is able to reduce absolute mean bias across the full region to zero, however this is achieved at the cost of introducing strong negative biases which cancel the remaining positive biases, as shown in the change in spatial bias from Fig. 4 a to Fig. 4 b. Since MBS uniformly subtracts bias from all sites across the region, areas of low positive bias in SNODAS (eg. along the US border) have their SWE estimates reduced too aggressively and now exhibit a strong negative bias. This subtraction process can lead to unphysical, negative estimates of SWE which should be discarded if this bias correction technique is to be used in practice. Additionally, areas with the strongest positive bias in SNODAS throughout the northern region have their SWE estimates reduced by too little and continue to display a strong positive bias.

MBS results in the creation of a SWE product that has been overcorrected in some areas and undercorrected in others, and leads to high RMSE in the final corrected dataset. Similar issues are also apparent temporally in the MBS corrected timeseries of Fig. 5, with an undercorrection of SWE in the years before 2015, and an overcorrection during 2015 and the years that follow. This residual error suggests that MBS is unable to fully capture spatio-temporal differences in the SNODAS bias and that more sophisticated techniques should be investigated.

### 3.2.2 Linear regression

A limitation of MBS is that it is unable to benefit from predictor relationships between the snow bias and climate variables. Using daily SNODAS SWE as a predictor, SLR displays skill in significantly reducing absolute mean bias. However, this technique seems to overcompensate in the correction of the strong bias in the northern region of the study area, especially during MAM where the absolute bias drops below zero to -3.2 mm SWE (Fig. 2). RMSE is reduced from the uncorrected SNODAS values to 15-20 mm SWE on average. Similar to MBS, the SLR corrected dataset also exhibits a RMSE difference between the northern and southern regions. We note the largest decreases in RMSE in the northern portion of the study area across all time periods with improvements of approximately 50% over the uncorrected SNODAS values. However, only slight reductions in RMSE are observed throughout the southern region.

SLR performance across the full spatio-temporal study range exhibits similar results to that of its partitioned comparison, with absolute mean bias reduced to approximately -1.25 mm SWE, and overall RMSE lowered by 45% (to 14.9 mm SWE) compared to that of the default SNODAS bias (Fig. 3). In order to determine whether the inclusion of additional predictors improves the performance of linear regression, MLR was also examined. When run with the predictor set described in Table 2,

MLR exhibits similar performance to SLR with approximately the same reductions in absolute bias and only slightly lower RMSE ($RMSE_{MLR} = 13.66$ mm SWE vs. $RMSE_{SLR} = 14.9$ mm SWE).

SLR continues to improve upon the results of MBS by further reducing absolute mean bias and RMSE at individual locations as shown in Fig. 4 c. The results of this technique show significant reductions in the spatial bias present in SNODAS. However, this technique also suffers from bias overcorrection. Since the SNODAS bias is not homogeneous across all snow survey sites,

areas of negative bias in SNODAS are corrected by the SLR model to be even more negative (as is seen at a set of survey sites in Fig. 4 c along the coasts of Lake Huron, Lake Ontario and Lake Superior). SLR improves the overall positive bias across the majority of the northern region sites, however a strong positive bias persists at many locations after SLR is applied, suggesting undercorrection at some locations. Similar to MBS, we note both an overcorrection and undercorrection of SWE in the timeseries of Fig. 5 (with a transition occurring again in 2014), confirming our assumptions that the linear regression

methods are unable to account for heterogeneity and nonstationarity in the bias between years.

### 3.3 Nonlinear methods

The DT method displays further improvements over MBS and SLR in terms of model skill, with the second lowest overall RMSE between 3-8 mm SWE on average, coupled with near-zero absolute mean bias when partitioned spatially and temporally (Fig. 2). Differences in RMSE are quite small between each region and time period, and the resulting RMSE between DT and

the in situ observations is substantially lower, on average, than that of uncorrected SNODAS (an 80% improvement). We note similar large improvements in model performance using the DT method across the full region for all months, with an overall RMSE of 4.03 mm SWE and absolute mean bias of 0.6 mm SWE.

Building on the improvements from DT, we find that RF displays the best overall skill of all tested models by producing SWE estimates with low absolute mean bias and the lowest overall RMSE when compared with in situ SWE. As noted in

the predictor importance scores of Table 2, RF incorporates information from a suite of predictor variables which allows the model to better understand how SWE biases change in both time and space. RF was found to consistently outperform the other models for all time periods for both northern and southern regions of our study area, as shown in the partitioned model run summary statistics in Fig. 2, with absolute bias reduced below 1 mm SWE and RMSE between 3 and 5 mm SWE. Furthermore, RF continues to outperform other methods of bias correction when the model is trained and run over the full spatio-temporal

domain, resulting in an RMSE of 3.64 mm SWE and absolute mean bias of only 0.2 mm SWE as shown in Fig. 3. This is an 86% reduction in RMSE compared to the uncorrected SNODAS RMSE and a significant improvement over the 45% reduction achieved by SLR and the 20% reduction in RMSE from MBS. Since the the RF is composed of an ensemble of DT models, it is not surprising that both methods perform similarly when run with the same predictor set, with RF slightly outperforming a single DT, because the ensemble is more robust and reduces systematic model error caused by overfitting.

As the bias in SNODAS is nonstationary (Fig.1b), we next evaluate the bias correction methods separately for a sub-period of high bias (2011-13), and one of low bias (2014-17). This test is performed using the same predictor variables in Table 2, excluding Year Id; i.e., we implicitly assume stationarity within each sub-period. During the high bias period (with a default bias of 27.9 mm SWE and default RMSE of 38.5), we find similar results to the full period. MBS, SLR and RF all reduce the absolute mean bias down to less than 1 mm SWE, and RF reduces RMSE to the lowest value of 2.7 mm SWE, compared to 5.7 and 26.6 for SLR and MBS, respectively. The low bias period (with a default bias of 9.28 mm SWE and default RMSE of 16.5), again exhibits a similar pattern in model performance to the full period, with all models reducing the absolute mean bias to less than 1 mm SWE and RF again showing the lowest RMSE of 4.9 mm SWE, compared to 14.0 and 13.6 for SLR and MBS. In summary, the sub-period analysis shows consistent performance from the RF model, but improved performance of the SLR model during the high bias period, when the bias in SNODAS appears more uniform from year-to-year (Fig.1b).

The RF model displays the best overall performance in terms of reducing bias and RMSE, and this skill is demonstrated spatially in Fig. 4 d. Compared to the other bias correction methods, the RF model is the most effective at reducing the spatial bias in SNODAS, with only small differences between model corrected SWE values and in situ SWE across the majority of the region. This accuracy is also evident in the timeseries of domain-averaged SWE values shown in Fig. 5, with the RF corrected SWE estimates closely tracking the in situ observations across all years. Comparisons of interannual correlations further emphasize the strengths of the nonlinear techniques over traditional bias correction methods at capturing changes in bias over time. Interannual correlations between RF corrected SWE and in situ are the highest at approximately 0.99, with correlations of approximately 0.93 for linear regression and of approximately 0.90 between the default SNODAS and in situ SWE. The RF model is therefore selected as the best-performing candidate model to perform bias correction on the Ontario-wide SNODAS data.

## 4   Application of the random forest model

In this section, we apply the trained RF model to the full 1 km SNODAS grid for all of Ontario (approx 1.5 million grid cells, Fig. 1 a), and derive a gridded estimate of corrected SWE throughout the entire region. This operation takes around 30 seconds per day of SNODAS observations (approximately 1.5 megabytes per day in storage space) on a modern, 4-core desktop computer. After running the RF model at 1 km resolution, we plot the resulting average monthly SWE bias between SNODAS and the RF corrected grid in Fig. 6 for December through May ($SNODAS - RF$). From these plots we note a strong positive monthly bias from January through April with the largest bias in SNODAS SWE estimates in March and April (averaging 57.7 mm SWE and 55.8 mm SWE, respectively), when the amount of snow on the ground is generally at its highest in Ontario. We also note a strong bias East (downwind) of Lake Superior and Lake Huron where SNODAS may be producing too much lake-effect snow. Through the application of the RF bias correction, estimated mean SWE during December to May in the study region outlined in Fig. 1 a is reduced by approximately 33 mm (Fig. 6).

## 4.1 Water balance analysis

The difference in snow water volume between uncorrected and bias-corrected SNODAS SWE has important implications for understanding the regional water balance of Ontario. The reduction in mean SWE resulting from the bias correction should reduce the regional melt estimates, which we estimate using hydrographs of area-normalized discharge from three Ontario river gauges for the period 2011 to 2018: Pic River (48.77° N) near Marathon, the South Branch Muskoka River (45.14° N) at Baysville, and the Thames River (42.54° N) at Thamesville (basins outlined in Fig. 1 a). Monthly melt water amounts are estimated as the (negative) SWE differences between consecutive monthly means from the SNODAS and RF SWE datasets (Erler et al., 2019). To compare the melt volumes with normalized discharge values, the melt is averaged over the drainage area associated with each stream gauge. We argue that this provides a reasonable estimate of the amount of water being released from the snowpack during the spring freshet period in each watershed. Note, however, that this does not include losses of water due to evapotranspiration or additional water input from rainfall.

Figure 7 shows the timeseries of area-normalized discharge and estimated melt rate over the study period for the three drainage areas. The timing of observed peak streamflow closely aligns with melt rate peaks during the spring freshet at the northern gauges of Pic River and the South Muskoka River. Since the melt water estimates do not include rainfall, they should be considered a conservative estimate of potential spring discharge.The melt volumes derived from the corrected dataset are close to, but mostly below, observed discharge values in the two northern catchments, while the estimates based on the uncorrected SNODAS SWE data significantly exceed the observed discharge, and can thus be considered unphysical. This serves as an independent validation of the physical plausibility of the bias correction method proposed here.

We further note that the differences between the corrected and uncorrected melt estimates are most apparent during the period of high bias prior to 2015. In this context it is also interesting to note that the accumulation of SWE in the uncorrected SNODAS dataset exceeds the total amount of precipitation (based on the NRCan dataset) for most winter months prior to 2015, and for isolated winter months after 2015. In the southern Thames River watershed, on the other hand, there exists a much lower bias between SNODAS and RF-corrected melt compared to the two northern watersheds, which is consistent with the previously discussed spatial pattern of biases in Fig 1 a. In addition, the Thames River watershed is not snowmelt dominated, so the biases do not affect streamflow in the same way as they do in the two northern watersheds. The changes in the magnitude of snowmelt shown in Fig. 7 suggest that the RF bias-corrected SWE constitutes an improvement over the uncorrected SNODAS-derived melt estimates throughout the study region, and that the RF-corrected dataset could provide a valuable new resource for hydrologic modeling and flood risk forecasting.

## 4.2 Discussion

Linear regression and machine learning techniques have previously been used effectively across the geosciences for bias correction of global and regional climate model output (Teutschbein and Seibert, 2012; Li et al., 2010; Lary et al., 2009; Reichstein et al., 2019; Shen, 2018). Previous studies on the estimation of North American SWE using artificial neural networks and support vector machines also exhibit similar results, with machine learning techniques outperforming general linear models

(Snauffer et al., 2018; Xue et al., 2018). However, recent work by Dixon et al. (2016) and Ehret et al. (2012) suggests that bias correction methods have their own associated uncertainties which must be considered when applied to datasets like SNODAS. These studies suggest that potential inconsistencies can exist between real-world and model dynamics, and their interactions with bias correction techniques. This can lead to unphysical changes in the relationships between variables and model dynamics, and even violate basic physical principles. This last point is relevant to this study, as some models (like MBS) over-correct SWE on the ground to negative values, which are physically meaningless. Our research has found that more sophisticated nonlinear statistical techniques like DT and RF produce bias-corrected SWE values that adhere more closely to these physical principles.

We must also consider uncertainties in the in situ snow survey data record. Hand measured SWE observations are generally considered to be of high accuracy; however, measurement error can still occur. Common issues include snow sticking to the inside of the measurement device or falling out of the bottom of the device due to improper soil capping (López-Moreno et al., 2013; ECCC, 2000). Issues like these can lead to underestimations in SWE when measurements are being recorded. Furthermore, from the available documentation by Metcalfe (2018), not all CAs use the same snow coring device and measurement techniques when retrieving SWE samples and this may result in systematic differences in their reported SWE estimates. Errors in the reference dataset can propagate through into the bias correction model during training and negatively impact the reliability of the model, even away from the snow survey locations. Additional error also arises in our comparison of point to grid data since our analysis assumes that the snow survey data is generally representative of the surrounding area in the containing 1 km SNODAS grid cell. While snow survey locations are selected to be representative of their surrounding landscapes Authorities (1985), snow density varies drastically over small spatial scales, and this assumption of homogeneity contributes to further uncertainty in our analysis (Molotch and Bales, 2005).

An additional source of uncertainty in the water balance analysis (section 4.1) arises from the lack of sublimation and evapotranspiration estimates as components of our melt calculation. Improving our estimates of melt and runoff in these basins requires additional high frequency precipitation data (including phase estimation), temperature data along with a series of non-obvious judgements to estimate sublimation (Dingman, 2015). Furthermore, the inclusion of a hydrologic or land surface model, which would be necessary to properly account for sublimation and evapotranspiration would also not be helpful for this purpose, as these models compute snowpack internally and one would be left with a comparison against modeled snowpack SWE. We also note that average liquid precipitation during the spring freshet exceeds the influence of potential evapotranspiration during this region at this time (Isabelle et al., 2020), and it can therefore be argued that snowmelt places a lower bound on the spring freshet volume (Erler et al., 2019). While the inclusion of these additional components to the water balance equation would improve melt estimates, through our comparison of SNODAS SWE differences with those estimated from NRCan climate normals and streamflow, it is clear that the uncorrected SNODAS values are unphysical, while the bias-corrected values appear reasonable.

Any additional uncertainties that exist in SNODAS and the NRCan gridded precipitation product (which are ingested as predictors into the bias correction models) will further contribute to the overall error in the bias corrected SWE dataset (Hay et al., 2006). Uncertainties in the SNODAS numerical forecast model along with measurement error from the datasets being

assimilated by each product add to the total uncertainty of the system. Furthermore, we note again that the reference period of the climate normals that have been used to characterize the climate in the RF model, is 1981 to 2010, while the study period is 2011 to 2018. This may introduce additional uncertainty due to decadal variability and transient shifts in climate; however, since only long-term averages (monthly normals) have been employed for this purpose, the error is likely small. Additionally,

a study by Sinha et al. (2019) suggests that RF is sensitive to the spacial representativeness of the bias throughout the region and that spatio autocorrelation in the training and testing datasets used in the RF may also negatively influence the accuracy of our model fit.

While there is no clearly documented reason behind the SNODAS SWE bias that occurred after 2015, we believe this may be the result of new datasets being inserted into the data assimilation scheme used by SNODAS. Although one may argue that

since the general magnitude of the SNODAS bias is reduced post 2014, a bias correction of SNODAS SWE in this region is unnecessary. We suggest that the bias correction is still a valuable contribution, since the SNODAS bias remains non-zero (approximately 5 mm SWE on average, and even higher throughout the northern region) during this period when compared with in situ, and the extended bias corrected data record allows us to better calibrate current hydrologic models. Another area of potential interest for other groups using SNODAS in Canada exists in the latitudinal gradient of bias we note in Fig. 4 a,

which suggests that the mean SNODAS bias increases in magnitude as we move further away from the US border (SNODAS is a US product which mainly ingests US data).

Each of the bias correction methods examined here shows skill in reducing the absolute bias present between SNODAS and in situ SWE observations, from the default 16 mm SWE in SNODAS to less than 1.5 mm SWE across all techniques. MBS and SLR exhibit an inability to capture year-to-year variability present in the bias and often overcorrect or undercorrect the amount

of SWE on ground, resulting in high RMSE between their corrected estimates and in situ observations. The more sophisticated machine learning techniques display further improvements in skill, with RF reducing RMSE by approximately 86% compared to that of the uncorrected SNODAS RMSE, and a reduction in absolute bias throughout the region to 0.2 mm SWE. The additional predictors combined with the ability of the model to capture nonlinear behavior, allows the RF model to closely reproduce observed SWE values and remain within physically plausible limits. The RF model also provides insights into the

strengths of the relationships between biases and various model predictors, suggesting a connection between SNODAS biases and elevation, total precipitation and air temperature. Furthermore, it is also evident that the bias diminishes over time, even though this may not be adequately reflected in the predictor importance ranking of the calendar year variable. Unfortunately, due to lack of documentation regarding changes in the assimilation system of SNODAS, it is not possible to identify the reasons behind these changes.

In this study we have only employed simple linear regression and decision tree-based methods of bias correction. Neverthe-less, we have demonstrated that nonlinear techniques can be used very effectively for bias correction, and are far superior to linear methods. Neural networks and support vector machines have also been effectively implemented for the purpose of bias correction in the geosciences (Lary et al., 2009). A paper by Xue et al. (2018) also found that machine learning methods can act as effective operators at estimating North American snow mass. It is possible that these other machine learning techniques

may offer further improvements to the methods examined here, and should be considered in additional followup work. Fur-

thermore, it has been suggested by Reichstein et al. (2019) and Shen (2018) that deep learning methods can provide powerful new perspectives in addressing common challenges in information extraction for water resource research. However, the region that was considered in this study is relatively small and climatologically homogeneous, and the number of in situ observations is likely insufficient to justify the use of more complex techniques that typically require very large training data sets. If, on the other hand, bias correction were to be attempted on a larger scale, for example the entire SNODAS domain, a more complex technique should be considered: likely a deep neural network, potentially with recurrent properties or convolutional layers, so as to account for memory effects and spatial structure. In this scenario, it would also be possible to make use of significantly more in situ observations across North America (e.g., SNOTEL sites), that could be used to train such a model.

## 5 Conclusion

Improving the quantification of southern Ontario SWE provides us with a foundation for better understanding regional flood risk along with more accurate forecasts of local water availability. Ontario's low density of in situ measurement stations results in large spatio-temporal observational gaps with unknown amounts of accumulating SWE. These observational gaps can be filled using snow model estimates or data assimilation techniques, however as noted in the case of SNODAS, there exist uncertainties and biases associated in these products. In this work, we demonstrate the skill of a variety of bias correction techniques and find that more sophisticated, nonlinear models offer enhancements in precision and accuracy to traditional statistical methods of bias correction. When applied to SNODAS, RF was found to reduce absolute mean bias across the region to less than 1 mm SWE and also displayed the strongest reduction in RMSE to less than 3 mm SWE (an RMSE improvement of 86% over default SNODAS). The result of this technique is an RF bias correction model that, when applied to SNODAS, offers improved daily estimates of SWE which can then be used as inputs to other forecasting systems and models. Examining melt estimates derived from area-averaged hydrographs over three basins within our study region revealed that the bias corrected SNODAS SWE data improved area melt estimates dramatically during the spring freshet when compared to the unphysical melt volumes exhibited by default SNODAS SWE. This improved skill in reducing bias from the nonlinear techniques suggests that machine learning techniques like RF (using similar predictor datasets as examined here) may also be applicable to future studies working to reduce SWE biases across other regions, in other gridded products. Through comparisons with in situ SWE measurements, nonlinear bias correction techniques improve the accuracy of SNODAS SWE estimates, and the resulting bias corrected dataset can therefore be used to further advance our understanding of the regional water balance and forecasting of future flood events.

*Data availability.* SNODAS SWE data is publicly available for download via National Snow and Ice Data Center (https://doi.org/10.7265/N5TB14TC). NRCan ANUSPLIN gridded products can be downloaded from Natural Resources Canada (https://cfs.nrcan.gc.ca/projects/3). ECCC snow survey records are available for public download on GitHub (https://github.com/frasertheking/ontario_snow_surveys).

*Author contributions.* AE and SF identified the bias and conceived the project. All authors developed the methodology and provided interpretations of the results. CF and AE supervised this work. AE obtained the data, FK organized the data, trained and tuned the models, developed the bias corrected dataset, and produced Figures 1-6. AE performed the water balance calculations and produced Figure 7. The introduction was written by SF, with the remainder of the document written by FK and reviewed by AE and CF.

5   *Competing interests.* The authors declare that they have no conflict of interest.

*Acknowledgements.* This study was supported by an Engage grant from the Natural Sciences and Engineering Research Council of Canada.

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

**Table 1.** Descriptions of the primary datasets used in our bias correction methods including relevant regression variables, their resolution, observational record coverage and data references.

| Dataset | Variable(s) | Horizontal resolution | Data period | Reference |
|---|---|---|---|---|
| Snow Surveys | Snow water equivalent | 383 points | Jan 1933–May 2018 | ECCC (2000) |
| SNODAS | Snow water equivalent, Total precip. | 1 km | Jan 2010–Dec 2018 | Carroll et al. (2001) |
| NRCan | 2-Meter temperature, Total precip. | 10 km | Jan 1979–Jan 2010 | McKenney et al. (2011) |
| Provincial DEM | Elevation | 30 m | May 1978–Mar 2018 | MNRF (2019) |

**Table 2.** Predictor names and details used in the decision tree, multiple linear regression and random forest bias correction models. Also included are their respective variable units, measurement timescales, data sources and variable importance scores produced by the random forest model.

| Predictor | Description | Units | Time scale | Data source(s) | RF Importance |
|---|---|---|---|---|---|
| SWE | SWE on ground | Millimeter | Daily | SNODAS | 0.68 |
| T2 | 2 meter air temperature | Celsius | Monthly | NRCan | 0.08 |
| TP Difference | NRCan - SNODAS total precipitation | Millimeter | Monthly | NRCan, SNODAS | 0.08 |
| Year Id | Year of observation | 1-of-c Indicator | – | – | 0.07 |
| Elevation | Height relative to sea level | Meter | – | Ontario Government | 0.06 |
| Month Id | Month of observation | 1-of-c Indicator | – | – | 0.01 |

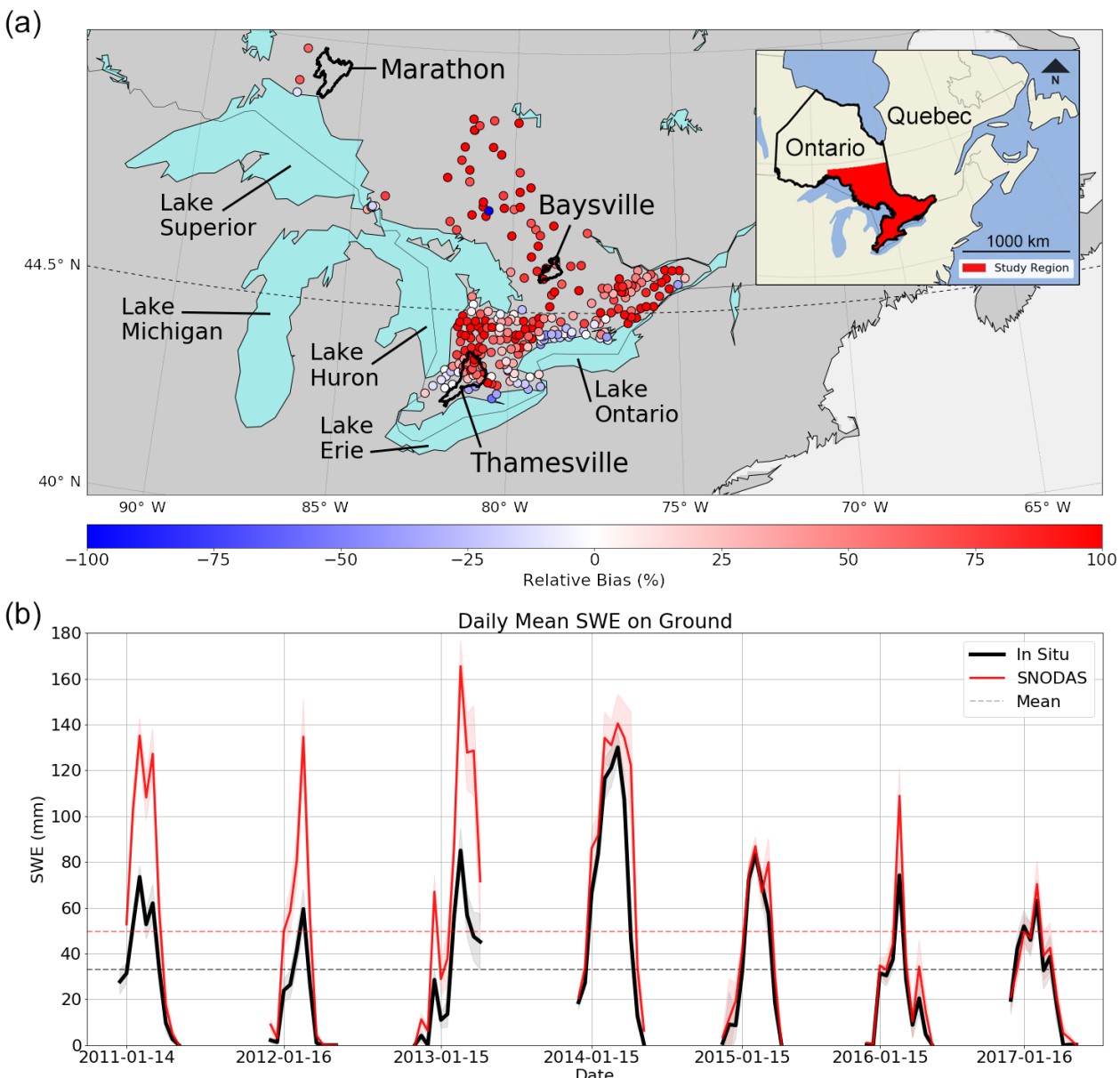

**Figure 1.** (a) Relative mean bias between SNODAS and in situ SWE aggregated for each snow survey site (colored points). Thicker black contours show the boundaries of the three drainage basins in the water balance analysis (Section 4.1). (b) Daily mean SWE on ground estimates from all in situ survey sites and SNODAS, taken biweekly from November to May [2011-2017] at 383 locations across Ontario.

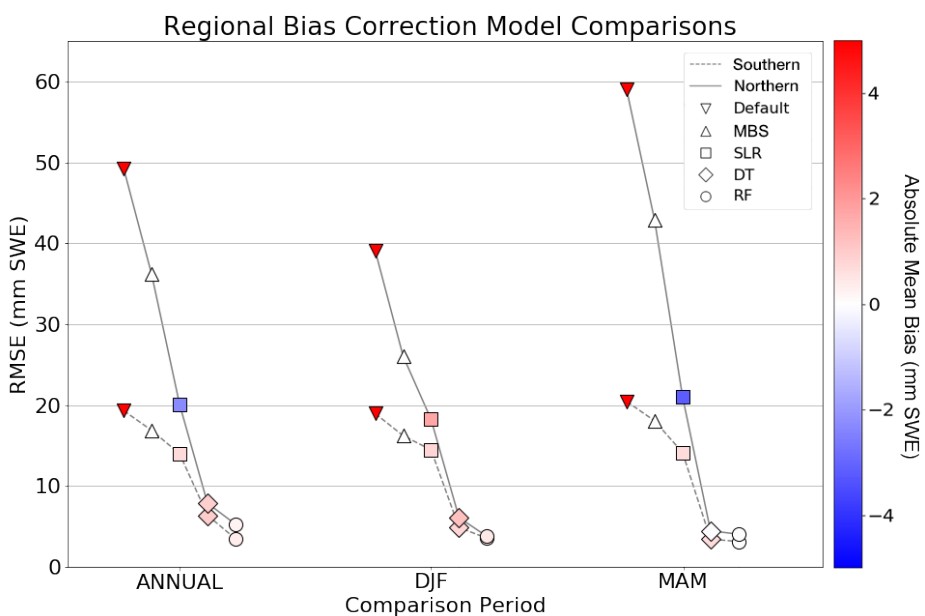

**Figure 2.** Performance results of regional bias correction methods (mean bias subtraction (MBS), simple linear regression (SLR), decision tree regression (DT) and random forest regression (RF)) for northern and southern geographic regions across DJF, MAM and the combined annual snow season (DJFMAM).

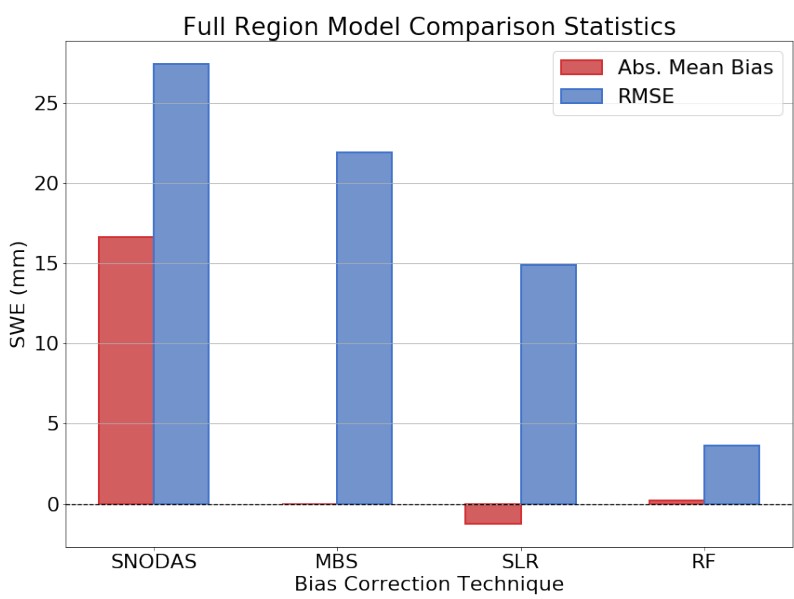

**Figure 3.** Bias correction model performance results for each technique across the full spatio-temporal domain.

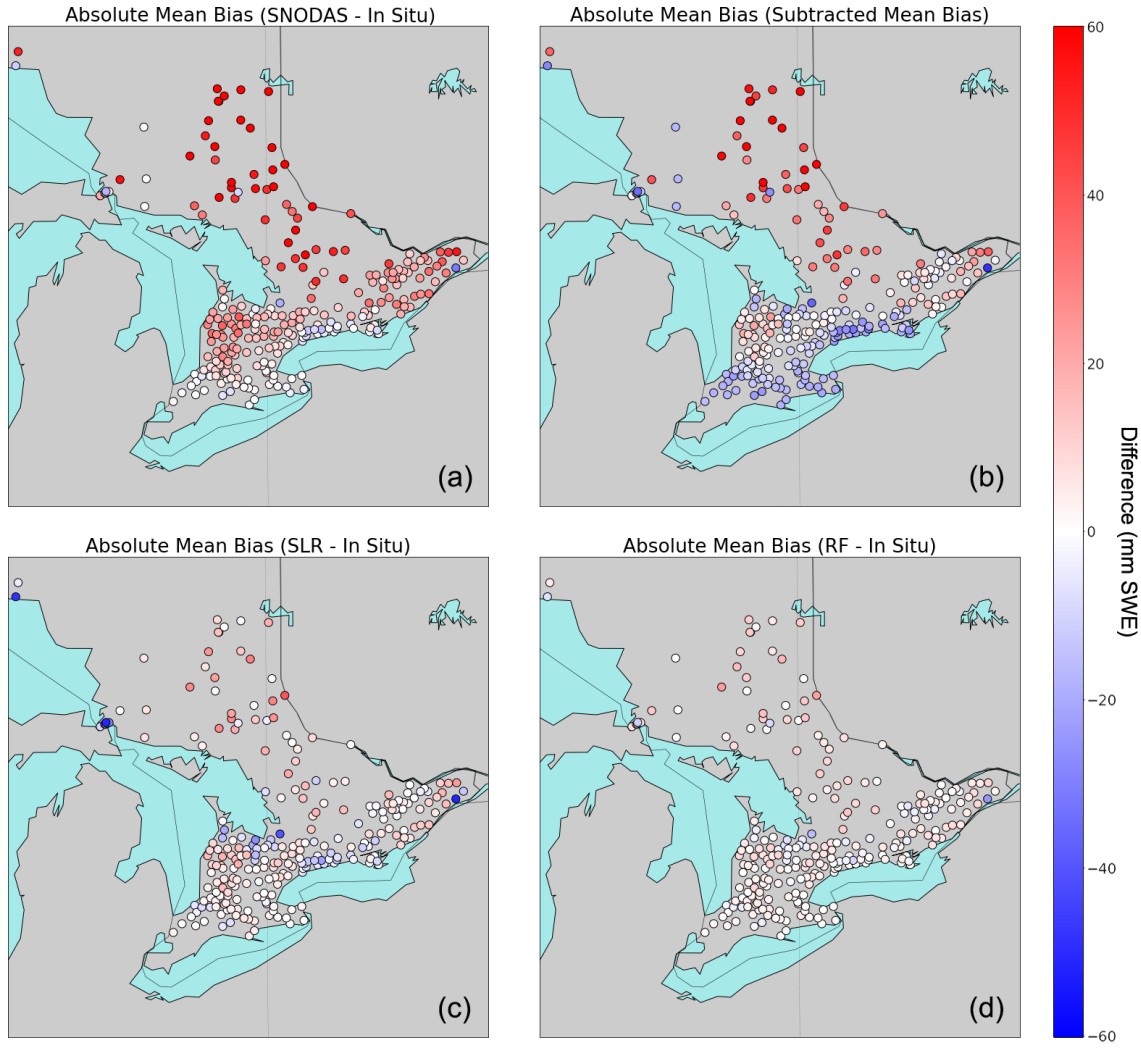

**Figure 4.** Absolute mean bias comparisons between in situ SWE and (a) SNODAS, (b) MBS, (c) SLR, and (d) RF, averaged at each snow survey site over the full study period.

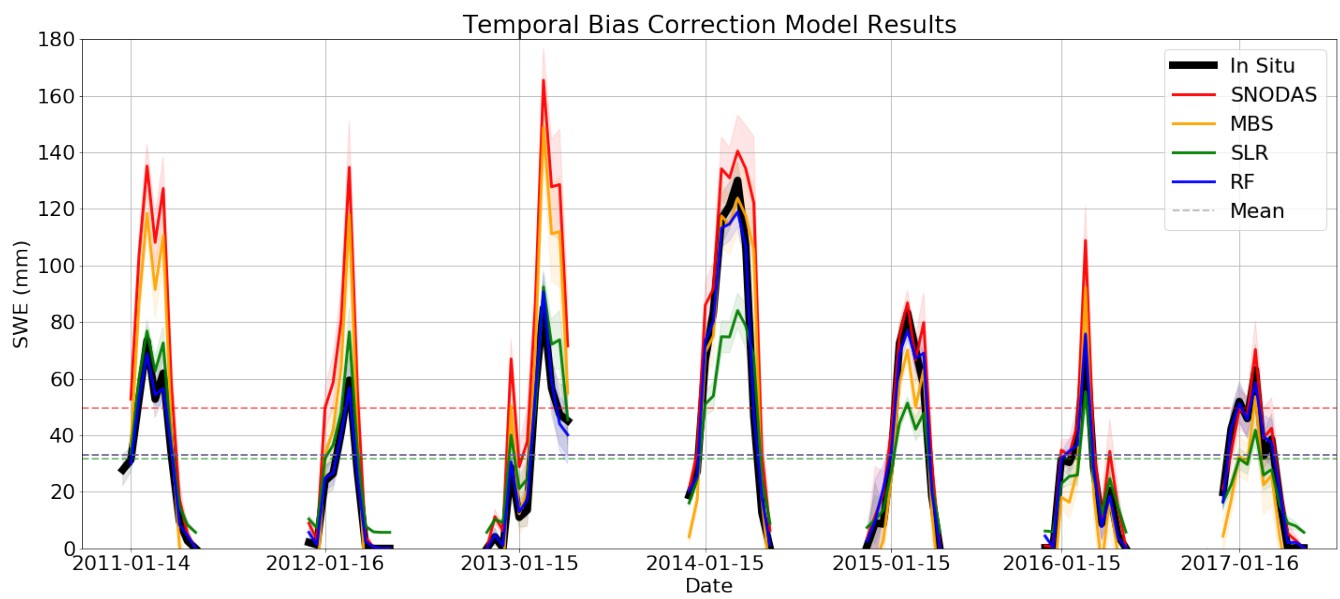

**Figure 5.** Daily mean SWE on ground for the MBS, SLR and RF bias corrected datasets, the default SNODAS SWE dataset and in situ SWE records. Shaded areas represent 95% confidence intervals based on the region data sample.

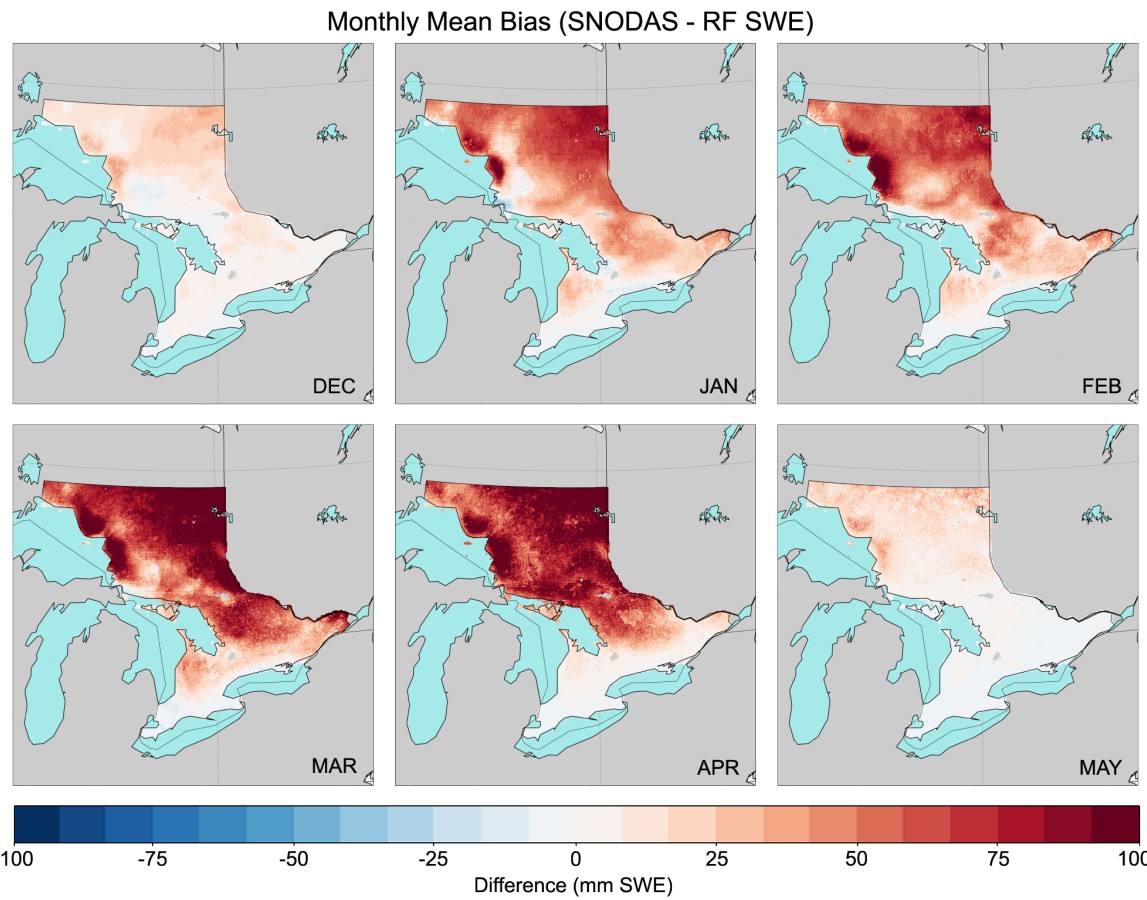

**Figure 6.** Monthly mean bias of SWE on ground between SNODAS and the RF bias corrected SWE dataset over December, January, February, March, April and May across the full study region at 1 km resolution.

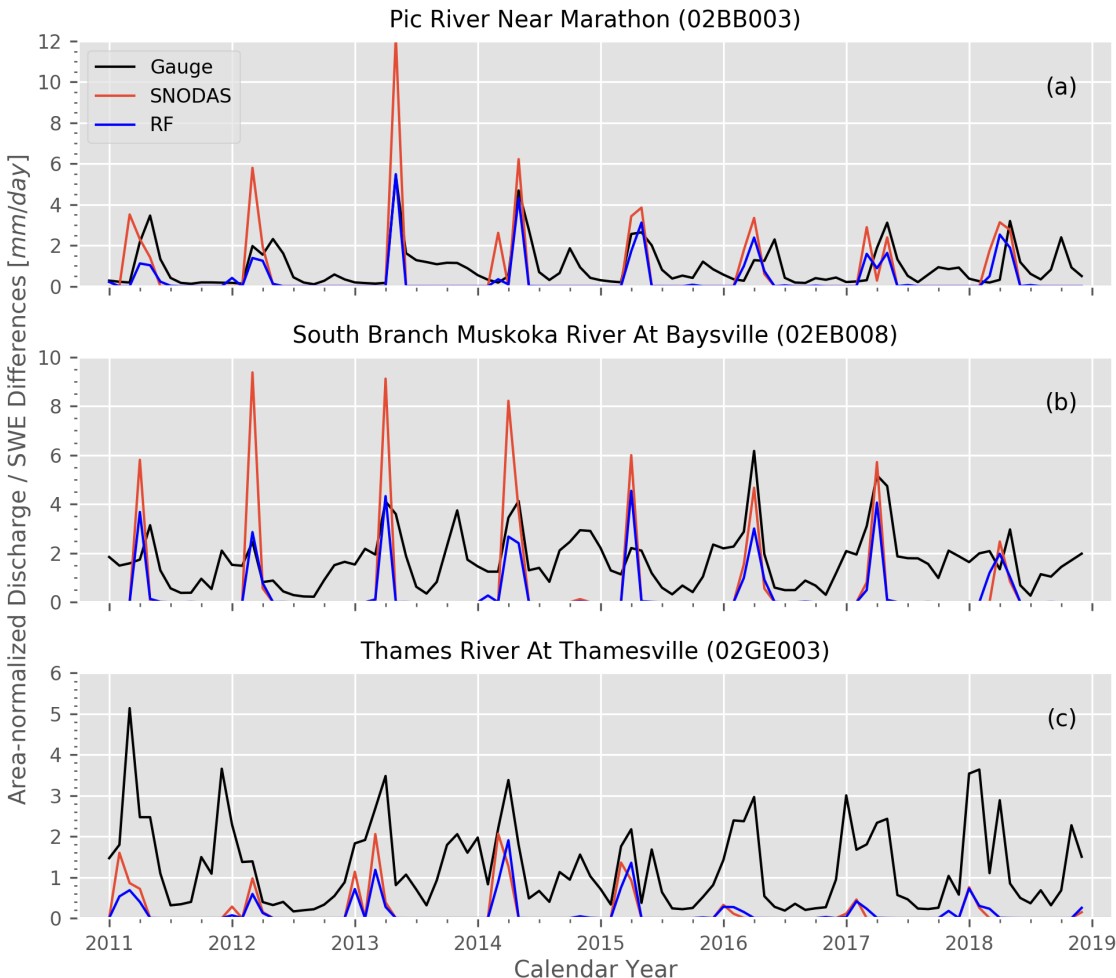

**Figure 7.** Monthly timeseries of area-normalized discharge from three river gauges in Ontario, along with corresponding melt estimates calculated from the SNODAS and RF corrected SWE datasets. Melt estimates are negative monthly SWE differences averaged over the drainage area of the corresponding gauge (see section 4.1).