# Peer review of "Application of machine learning techniques for regional bias correction of SWE estimates in Ontario, Canada"

_Hydrology and Earth System Sciences, 2019_

## Referee Comment (RC1) · Anonymous Referee #1 · 18 Feb 2020

General comments

This study quantifies bias between SNOWDAS assimilation dataset and in situ SWE observations in Ontario region (Canada) and compares efficiency of three different bias correction methods in terms of improvement of SWE prediction and estimated snowmelt volumes. The results indicate that there is a bias between SNODAS and in situ SWE, particularly in the period 2011-2013 and that the machine learning technique (random forest) approach outperforms simple mean subtraction and linear regression bias correction methods.

Overall, the manuscript is clearly written and has a good structure. The topic is relevant

and within the scope of the journal. I would like to make only a few general comments:

1) The results indicate that there is a clear difference in SNOWDAS agreement (against in situ SWE) in the period 2011-2013 and 2014-onwards. It will be interesting to see/understand why? Is it the change in assimilation frequency, sources used in assimilation, their accuracy? I think such understanding can then support the selection of approach used for bias correction. It has some implications also for the design of this study. If there is a step change in SNOWDAS, then it is not surprising that simple mean subtraction method is not working well for the entire period. It will be interesting to see why does the random forest outperform the other methods in such case and what factors are controlling its efficiency? (Is it because using year of observation?) Will it be not more fair in this case to compare the methods in two separate periods?

2) I think that the referencing (used in the Introduction and Discussion) can be improved. There are some relevant papers which are not addressed: e.g. Zahmatkesh et al. (2019) evaluating bias correction of SNODAS in Canadian basins or some studies cited in Lv et al. (2019) focusing on the accuracy assessment of SNODAS. Please consider to formulate how does this study compare to these studies (in Intro and Discussion sections).

3) I have to say that the part related to evaluation of the impacts of different bias corrected SWE estimates on snowmelt is not clear to me. Using monthly estimates without accounting for evapotranspiration and other processes is somewhat less robust. Comparison of observed daily discharge with daily simulations driven by a hydrologic model will be more representative example.

4) How to account for scale gap between SNODAS and in situ observations?

Specific comments

Fig.1b. What do the lines represent? Mean over 383 stations?

Fig.2,3,4,5. Please explain the meaning of abbreviations MBS, SLR, etc. in figure

caption.

References:

Zahra Zahmatkesh, Dominique Tapsoba, James Leach & Paulin Coulibaly (2019) Evaluation and bias correction of SNODAS snow water equivalent (SWE) for streamflow simulation in eastern Canadian basins, Hydrological Sciences Journal, 64:13, 1541-1555, DOI: 10.1080/02626667.2019.1660780

Zhibang Lv John W. Pomeroy Xing Fang (2019) Evaluation of SNODAS Snow Water Equivalent in Western Canada and Assimilation Into a Cold Region Hydrological Model, WRR, https://doi.org/10.1029/2019WR025333

―――――――――――――――――

---

## Referee Comment (RC2) · Anonymous Referee #2 · 5 Mar 2020

This very interesting paper of King, et al. compares different methods for reducing the bias between in situ measurements of SWE and the gridded SNODAS estimates for the region of Ontario. The correction methods include simple mean bias subtraction, linear regression and machine learning methods. The paper is very well written and it is worth to be published after some minor changes. Some comments and recommendations:

First of all and most important the applied machine learning methods are not described at all and references are missing. I don't think that all readers of this journal are familiar with Decision Trees (DT) and Random Forest (RF) methods. Therefore a short description should be included, especially explaining the RF model in more detail, which shows

the best results, and what's the difference to the DT models. Related to that comment, it doesn't make too much sense to mention on page 5 (line 30) that you run the model with a forest size of 100 trees and tree depth of 15, when you don't explain what that parameter mean.

Additionally, there are some points which are not clear to me and which should be clarified before publishing the paper: You didn't explain how you handled the scaling issue when you compare point data and gridded data (up- or downscaling?). Since you could identify a change in the bias between the first and the second half of the period, it would be reasonable to split the analysis into these two periods and fit different models and take 2 different means separately for each period.

On page 3 you specify the 383 locations with in situ measurements. In line 14-15 you write that an average SWE is estimated taken from 10 fixed sampling stations. What does this mean? Is this the average SWE for Ontario estimated from 10 stations, or is this the average for each of the 383 stations taken from the 10 surrounding stations?? Page 5: You should mention that the period of 1981-2010 is used for calculating the climatology, which is not clear. Also, you should explain why you have used the difference between the precipitation estimates from NRCAN and the SNODAS! It would be interesting to see the results if you would include actual meteorological observations as predictors (for example available at: https://data.noaa.gov/dataset/dataset/global-surface-summary-of-the-day-gsod, provided by the National Centers for Environment Information). I could imagine that in that case the importance of these variables would not be neglectable and could further improve the bias correction.

Page 7: When you write in 3.2.1 about mean bias, I suppose that this mean bias is calculated as the average of the mean bias of all stations? Similar to that I'm a bit confused about what you write on page 8 regarding SLR. I was assuming that you fit a regression model for each station individually. But that seems to be not the case, otherwise I could not understand why there should be a bias overcorrection. It would be nice if you could clarify this, whether you fitted separate models for each station

or not. Although you wrote in the beginning that you took 75% for training, you didn't mention if all the calculated verification measures refer to the remaining 25% testing period.

In the legend of Figure 2 you write Lower and Upper. Shouldn't it be southern and northern?

---

## Author Comment (AC1) · 3 Apr 2020

Reviewer 1

General Comment:

This study quantifies bias between SNOWDAS assimilation dataset and in situ SWE observations in Ontario region (Canada) and compares efficiency of three different bias correction methods in terms of improvement of SWE prediction and estimated snowmelt volumes. The results indicate that there is a bias between SNODAS and in situ SWE, particularly in the period 2011-2013 and that the machine learning technique

(random forest) approach outperforms simple mean subtraction and linear regression bias correction methods. Overall, the manuscript is clearly written and has a good structure. The topic is relevant and within the scope of the journal. I would like to make only a few general comments.

General Comment Response:

We thank the reviewer for their comments, and we will work to incorporate their suggestions to improve our currently submitted manuscript. Our responses to each of the reviewer's questions/comments is included below.

Specific Comment 1:

The results indicate that there is a clear difference in SNOWDAS agreement (against in situ SWE) in the period 2011-2013 and 2014-onwards. It will be interesting to see/understand why? Is it the change in assimilation frequency, sources used in as simulation, their accuracy? I think such understanding can then support the selection of approach used for bias correction. It has some implications also for the design of this study. If there is a step change in SNOWDAS, then it is not surprising that simple mean subtraction method is not working well for the entire period. It will be interesting to see why does the random forest outperform the other methods in such case and what factors are controlling its efficiency? (Is it because using year of observation?) Will it be not more fair in this case to compare the methods in two separate periods?

Specific Response 1:

We agree with the reviewer that the change in bias post-2014 is of interest, and we mention on lines 27-30 of section 4.2 that newly assimilated datasets are likely the dominant contributing factors to the reduction in the intensity of the SNODAS bias during this period. We argue that while the bias is reduced post-2014, it is still non-zero and the approaches explored in our work continue to provide improvements to SNODAS estimates during this time. The decision tree and random forest approaches

outperform traditional methods like SLR and mean bias subtraction due to this nonlinearity in the bias and the ability for the machine learning techniques to recognize these patterns and better correct for them. As shown in the predictor importance scores of table 2, year does play a somewhat important factor along with other climatic variables like temperature and total precipitation. We agree with the reviewer that further descriptions of bias correction model performance (with respect to bias and RMSE) when trained/tested over these two separate periods (before and after 2014) would be beneficial, and therefore additional text describing the results of these comparisons has been added to the manuscript in section 3.3.

Specific Comment 2:

I think that the referencing (used in the Introduction and Discussion) can be improved. There are some relevant papers which are not addressed: e.g. Zahmatkesh et al. (2019) evaluating bias correction of SNODAS in Canadian basins or some studies cited in Lv et al. (2019) focusing on the accuracy assessment of SNODAS. Please consider to formulate how does this study compare to these studies (in Intro and Discussion sections).

Specific Response 2:

We thank the reviewer for recommending these relevant papers from Zahmatkesh et al. (2019) and Lv et al. (2019). These references have been added in the manuscript as additional motivation to our work in section 1 and section 4.2.

Specific Comment 3:

I have to say that the part related to evaluation of the impacts of different bias corrected SWE estimates on snowmelt is not clear to me. Using monthly estimates without accounting for evapotranspiration and other processes is somewhat less robust. Comparison of observed daily discharge with daily simulations driven by a hydrologic model will be more representative example.

Specific Response 3:

We thank the reviewer for this comment, as this point may not be immediately obvious: a direct comparison between SWE estimates and streamflow is not straight forward and presents a major methodological challenge, as outlined below. We will add additional discussion regarding the relationship between SWE, snowmelt, runoff and water balance estimates in section 4.1. The primary purpose of this section (and Figure 7) is to demonstrate that SNODAS SWE values are clearly too high and unphysical, especially during the time period before 2015, where estimated snowmelt exceeds total spring runoff in several cases. After bias-correction this is not the case anymore, suggesting that the bias-corrected values are at least plausible. The methodological challenge preventing direct validation of SWE estimates against streamflow gauges is the fact that runoff is generated by snowmelt and snowmelt has to be estimated from SWE changes. However, SWE also changes due to snow fall (and sublimation); snow fall, sublimation and melt occurring during the same time period cannot be separated easily (and can cancel each other). A better estimate of melt and runoff therefore would require additional data on precipitation, precipitation phase and/or temperature at high temporal frequency and a series of non-obvious judgements (such as estimating sublimation) would be required. This could be a topic of a potential follow-up study but is beyond the scope of this manuscript. A hydrologic or land surface model, which would be necessary to properly account for sublimation and evapotranspiration would not be helpful for this purpose, as these models compute snowpack internally and one would be left with a comparison against modeled snowpack (SWE). Furthermore, if SWE values from SNODAS were to be assimilated into the model, melt and runoff values would potentially be worse, since data assimilation violates mass conservation. As a case in point, we note that SNODAS also computes snowmelt internally, however, these values suffer from biases even larger than the biases in SWE. The reason for this is likely that snowmelt is not assimilated and at the same time artifacts are introduced by the assimilation of other variables (mass conservation is violated). Unfortunately, direct observation of snowmelt is not possible.

Specific Comment 4:

How to account for scale gap between SNODAS and in situ observations?

Specific Response 4:

In our analysis, we compare gridded estimates of SWE from SNODAS (1 km resolution) to snow survey estimates (which is essentially point data taken over 10 m). Due to the relatively high spatial resolution of SNODAS, along with the fact that the in situ measurement sites are taken at distances > 1 km from each other, we do not compare multiple in situ points to a single grid cell. This allows us to complete a simple point to grid cell comparison where we assume the snow survey SWE estimate is representative of the wider, containing grid cell. This assumption of representativeness across the grid cell introduces additional uncertainty, as SWE is highly variable at even small spatial scales, and we have therefore included additional details in the paper to make these uncertainties clearer to the reader in section 4.2.

Specific Comment 5:

Fig.1b. What do the lines represent? Mean over 383 stations?

Specific Response 5:

The reviewer is correct, the lines in Figure 1.b represent the daily mean SWE on ground for all survey locations (383 sites) across the full study period.

Specific Comment 6:

Fig.2,3,4,5. Please explain the meaning of abbreviations MBS, SLR, etc. in figure caption.

Specific Response 6:

We thank the reviewer for this comment, and we have included an additional description of the abbreviations for MBS, SLR, DT and RF in the caption of Figure 2.

References:

Lv, Z., Pomeroy, J. W., & Fang, X. (2019). Evaluation of SNODAS Snow Water Equivalent in Western Canada and Assimilation Into a Cold Region Hydrological Model. Water Resources Research, 55(12), 11166–11187. https://doi.org/10.1029/2019WR025333

Zahmatkesh, Z., Tapsoba, D., Leach, J., & Coulibaly, P. (2019). Evaluation and bias correction of SNODAS snow water equivalent (SWE) for streamflow simulation in eastern Canadian basins. Hydrological Sciences Journal, 64(13), 1541–1555. https://doi.org/10.1080/02626667.2019.1660780
* * *

---

## Author Comment (AC2) · 3 Apr 2020

Reviewer 2

General Comment:

This very interesting paper of King, et al. compares different methods for reducing the bias between in situ measurements of SWE and the gridded SNODAS estimates for the region of Ontario. The correction methods include simple mean bias subtraction, linear regression and machine learning methods. The paper is very well written and it is worth to be published after some minor changes. Some comments and recommendations:

[Figure]

General Comment Response:

We thank the reviewer for their comments, and we will work to incorporate their suggested changes to improve our currently submitted manuscript. Our responses to each of the reviewer's questions/comments is included below.

Specific Comment 1:

First of all and most important the applied machine learning methods are not described at all and references are missing. I don't think that all readers of this journal are familiar with Decision Trees (DT) and Random Forest (RF) methods. Therefore a short description should be included, especially explaining the RF model in more detail, which shows the best results, and what's the difference to the DT models. Related to that comment, it doesn't make too much sense to mention on page 5 (line 30) that you run the model with a forest size of 100 trees and tree depth of 15, when you don't explain what that parameter mean.

Specific Response 1:

We thank the reviewer for this comment, and agree that additional details should be included in the text which further describe the methodology behind the decision tree (DT) and random forest (RF) techniques we employ in this work. We have updated the document to include further references/details regarding what these techniques are and how they operate, along with further descriptions of what parameters like forest size and tree depth mean with respect to the RF model in section 2.3 of the manuscript.

Specific Comment 2:

Additionally, there are some points which are not clear to me and which should be comment clarified before publishing the paper: You didn't explain how you handled the scaling issue when you compare point data and gridded data (up- or downscaling?). Since you could identify a change in the bias between the first and the second half of the period, it would be reasonable to split the analysis into these two periods and fit

different models and take 2 different means separately for each period.

Specific Response 2:

In our analysis, we compare gridded estimates of SWE from SNODAS (1 km resolution) to snow survey estimates (which is essentially point data taken over 10 m). Due to the relatively high spatial resolution of SNODAS, along with the fact that the in situ measurement sites are taken at distances > 1 km from each other, we do not compare multiple in situ points to a single grid cell. This allows us to complete a simple point to grid cell comparison where we assume the snow survey SWE estimate is representative of the wider, containing grid cell. The snow survey sites are selected to generally be representative of the area around it and are not just random point measurements which would contain higher variability in their estimates. However, this assumption of representativeness across the grid cell introduces additional uncertainty, as SWE is highly variable at even small spatial scales, and we have therefore included additional details in the paper to make these uncertainties clearer to the reader. Furthermore, we agree with the reviewer that due to the change in the intensity of the bias post-2014, a description of how the bias correction models perform over these separate two periods would be interesting and complimentary to our analysis. Therefore, we have also updated the results section 3.3 of the paper with the results of this test.

Specific Comment 3:

On page 3 you specify the 383 locations with in situ measurements. In line 14-15 you write that an average SWE is estimated taken from 10 fixed sampling stations. What does this mean? Is this the average SWE for Ontario estimated from 10 stations, or is this the average for each of the 383 stations taken from the 10 surrounding stations??

Specific Response 3:

What we are referring to on page 3 is the method by which in situ measurements are retrieved (snow survey), where a sampling location is selected and then 10 point

measurements are taken using a snow coring device over approximately 10 meters at that location. These 10 SWE measurements along the snow survey are then averaged together to provide a single SWE estimate for that location. This is the technique used at all 383 in situ measurement sites. We now include additional details on how these measurements are retrieved to add further clarity to the reader in section 2.1 of the manuscript.

Specific Comment 4:

Page 5: You should mention that the period of 1981-2010 is used for calculating the climatology, which is not clear.

Specific Response 4:

We thank the reviewer for noticing this detail, and we now make the temporal period used for the calculation clear in the paper on page 5 (section 2.2.2).

Specific Comment 5:

Also, you should explain why you have used the difference between the precipitation estimates from NRCAN and the SNODAS! It would be interesting to see the results if you would include actual meteorological observations as predictors (for example available at: https://data.noaa.gov/dataset/dataset/globalsurface-summary-of-the-day-gsod, provided by the National Centers for Environment Information). I could imagine that in that case the importance of these variables would not be neglectable and could further improve the bias correction.

Specific Response 5:

We thank the reviewer for the comment; we have also considered this option; however, we have chosen to limit meteorological data to basic monthly climate normals. Analogous to the choice of model complexity, there is always a trade-off between accuracy, complexity and the risk of over-fitting. Using a large set of predictors requires a more complex model, which increases the risk of over-fitting. Therefore we have chosen to

only include monthly normal surface temperature and precipitation, as these two variables are usually readily available and characterize the type of climate reasonably well. The rational behind including climate variables was that, on average, snow characteristics (like density, albedo, ice content) vary between different climates. It is true that these characteristics would be predictable (to some extend) from the actual evolution of these meteorological forcings; however, the processes that govern such characteristics are very complex and involve long-term memory effects, which would require a much more complex model (like an LSTM), which would approach the complexity of physical snow models. Considering the data requirements and complexity of this approach, we believe that the use of monthly normals represents the best compromise. As for the reason, the difference between SNODAS average precipitation and NRCan normals was used, rather than total precipitation from NRCan (or SNODAS): this choice was made because notable biases in the precipitation fields used by SNODAS over Canada were found early on in the analysis, and it appears obvious that the size of these biases would have a first-order effect on the resulting SWE bias in SNODAS. At the same time, in order to reduce the number of input variables, we did not want to include multiple, possibly redundant, precipitation variables.

Specific Comment 6:

Page 7: When you write in 3.2.1 about mean bias, I suppose that this mean bias is calculated as the average of the mean bias of all stations? Similar to that I'm a bit confused about what you write on page 8 regarding SLR. I was assuming that you fit a regression model for each station individually. But that seems to be not the case, otherwise I could not understand why there should be a bias overcorrection. It would be nice if you could clarify this, whether you fitted separate models for each station or not.

Specific Response 6:

The reviewer is correct in that the mean bias is calculated as the difference between

the average SWE across the full temporal period for SNODAS minus the average SWE for all 383 survey sites (ie. the two dashed lines in the timeseries Figure 1.b). The reviewer is also correct in that we did not fit a SLR model to each station, but instead trained a single model across all survey sites for our full temporal period (as well as the partitioned upper and lower regions in Figure 2 to see if multiple models showed improvement; and we found they did not). The bias overcorrection in the linear techniques like SLR stems from the fact that the SLR is attempting to model a linear relationship across all years which is problematic due to the nonlinearity in the bias introduced post-2014. This results in an overcorrection in some periods and an undercorrection in others.

Specific Comment 7:

Although you wrote in the beginning that you took 75% for training, you didn't mention if all the calculated verification measures refer to the remaining 25% testing period.

Specific Response 7:

The reviewer is correct that the calculated verification measurements on model performance when performing validation testing refer to the remaining 25% of the dataset, we now make this clearer in the text in section 3.3.

Specific Comment 8:

In the legend of Figure 2 you write Lower and Upper. Shouldn't it be southern and northern?

Specific Response 8:

We thank the reviewer for noticing this naming discrepancy and we have updated the Figure 2 legend to show Southern and Northern instead of Lower and Upper.

---

## Referee Comment (RC3) · Anonymous Referee #3 · 14 Apr 2020

This work evaluates several bias-correction methods (simple subtraction, single and multiple linear regression, decision trees, and random forests) to SNODAS, resulting in a new data product that shows improved fidelity to in situ observations. The authors further develop a simple water balance analysis that exhibits the improved consistency of the inferred melt of the corrected model to streamflow observations. This work represents important progress to advancing the application of machine learning to water resources management in regions of snowmelt-dominated streamflow regimes.

Comments:

The potential strengths of machine learning are highlighted but a justification for the

selection of random forests (RF) is not particularly apparent. The authors mention applications of support vector machines and neural networks in geosciences detailed in Lary et al., 2009, a study of aerosol optical depth, but neglect to review specific literature around machine learning applications in SWE estimation (e.g. Wrzesien et al., 2017, Snauffer et al., 2018, Xue et al., 2018). A review of such advances is warranted.

RF model structure and hyperparameter descriptions should be moved to the methods section. The authors mention RF is run with a forest size of 100 and maximum tree depth of 15, but it is unclear how these hyperparameters were selected beyond a mention of "sensitivity tuning experiments". Generally hyperparameters should be tuned using a standard method (e.g. grid search, particle swarm optimization, evolutionary strategy, etc.) on each test split and reported accordingly. Is the maximum number of terminal nodes for a given tree specified or are the trees allowed to grow to full extent?

RF and DT are stated to be trained on 75% of the data and evaluated on the remaining 25% test set, but are also evaluated using a 10-fold cross-validation, resulting in an average RMSE reduction of 4.7 mm. The change to bias is unclear, as is the motivation for using both a 75-25 and 10-fold split structure. Since you've appropriately gone to the effort to run a full 10-fold cross-validation, why aren't you just using these results?

The manuscript would be strengthened with a description of the efforts you've undertaken to mitigate temporal and spatial auto-correlation in your training and test sets.

The manuscript would be strengthened with further descriptions of the efforts you've undertaken to mitigate overfitting. A comparison of training and validation errors would be an appropriate way to do this.

In Table 2, what are Year Id and Month Id? Are you using straight numerical values, cyclical temporal sin-cos pairs, 1-of-c indicators (Bishop, 1995)?

The water balance analysis averages melt over a watershed associated with a given stream gauge, asserting the stream gauge provides a reasonable estimate of snowmelt

while at the same time neglecting evapotransportation and rainfall (actually any precipitation). Such an assertion requires that evapotransportation and subsequent precipitation are not as significant a signal as snowmelt to runoff. This may be true, but it should be backed up by analysis and references, or minimally one of these. Baseflow should also be at a minimum mentioned.

You conclude that MBS and SLR exhibit an inability to capture year-to-year variability present in the bias, but interannual correlations are not present in the analysis. The ability of bias-correction methods particularly of the non-linear flavor to capture changes over time is arguably one of their greatest strengths, as simple offsets are more easily calculated, as you have done. A simple correlation calculation may serve as further evidence of the utility of the nonlinear method.

Fig 5 is hard to read with the scales and lines used, especially the in situ values, which are key to the plot. No description of shading used is given in the figure caption. Suggest changing line thicknesses/colors and/or adjusting scales, orientation, or paneling to make better use of available space.

References:

Bishop CM, 1995. Neural Networks for Pattern Recognition, Oxford University Press.

Snauffer AM, Hsieh WW, Cannon, AJ, Schnorbus, MA, 2018. Improving gridded snow water equivalent products in British Columbia, Canada: multi-source data fusion by neural network models. The Cryosphere, 12(3), 891-905.

Wrzesien ML, Durand MT, Pavelsky TM, Howat IM, Margulis SA, Huning LS, 2017. Comparison of methods to estimate snow water equivalent at the mountainrange scale: A case study of the California Sierra Nevada. Journal of Hydrometeorology, 18(4), 1101-1119.

Xue Y, Forman BA, Reichle RH, 2018. Estimating snow mass in North America through assimilation of AMSR-E brightness temperature observations using the Catchment

land surface model and support vector machines. Water Resources Research, 54(9), p.6488.

---

## Author Comment (AC3) · 28 Apr 2020

Reviewer 3

General Comment:

This work evaluates several bias-correction methods (simple subtraction, single and multiple linear regression, decision trees, and random forests) to SNODAS, resulting in a new data product that shows improved fidelity to in situ observations. The authors further develop a simple water balance analysis that exhibits the improved consistency of the inferred melt of the corrected model to streamflow observations. This work rep-

resents important progress to advancing the application of machine learning to water resources management in regions of snowmelt-dominated streamflow regimes.

General Comment Response:

We thank the reviewer for their comments and suggestions for improving the manuscript, and we will work to incorporate these changes into the article. Our responses to each of the reviewer's questions/comments is included below.

Specific Comment 1:

The potential strengths of machine learning are highlighted but a justification for the selection of random forests (RF) is not particularly apparent. The authors mention applications of support vector machines and neural networks in geosciences detailed in Lary et al., 2009, a study of aerosol optical depth, but neglect to review specific literature around machine learning applications in SWE estimation (e.g. Wrzesien et al., 2017, Snauffer et al., 2018, Xue et al., 2018). A review of such advances is warranted.

Specific Response 1:

We thank the reviewer for their suggestion to include additional motivation behind our selection of the random forest technique for bias correction. As mentioned by the reviewer, this choice primarily stems from the strengths this technique has shown in previous literature for bias correcting data in the geosciences (Reichstein et al., 2019; Shen, 2018; Lary et al., 2009). However, we agree that additional motivation with respect to bias correcting SWE would be beneficial, and we have now included additional literature focusing on the application of random forest bias correction towards SWE datasets from Wrzesien et al., 2017, Snauffer et al., 2018, Xue et al., 2018, Zahmatkesh et al., 2019 and Lv et al., 2019, in section 1 and section 4.2 of the manuscript.

Specific Comment 2:

RF model structure and hyperparameter descriptions should be moved to the methods section. The authors mention RF is run with a forest size of 100 and maximum tree

depth of 15, but it is unclear how these hyperparameters were selected beyond a mention of "sensitivity tuning experiments". Generally hyperparameters should be tuned using a standard method (e.g. grid search, particle swarm optimization, evolutionary strategy, etc.) on each test split and reported accordingly. Is the maximum number of terminal nodes for a given tree specified or are the trees allowed to grow to full extent?

Specific Response 2:

We thank the reviewer for this comment and question. During the model training phase of our analysis, we experimented with a variety of values for forest size and maximum tree depth to find a balance between model accuracy and run time efficiency. This sensitivity experiment was performed through a brute-force grid search approach of nudging each parameter value to find a set of parameters which exhibit both high general performance (low RMSE and bias), and an efficient RF model runtime. This test resulted in the selection of a forest size of 100, along with a max tree depth of 15. As per the maximum number of leaf nodes for each tree, this was left to allow each tree to grow to its full extent. We have moved some of the general model structure details (along with the hyperparameter descriptions) into the methods section and have also included further details on how the hyperparamaterization was performed in the same section (2.3) to add further clarity.

Specific Comment 3:

RF and DT are stated to be trained on 75% of the data and evaluated on the remaining 25% test set, but are also evaluated using a 10-fold cross-validation, resulting in an average RMSE reduction of 4.7 mm. The change to bias is unclear, as is the motivation for using both a 75-25 and 10-fold split structure. Since you've appropriately gone to the effort to run a full 10-fold cross-validation, why aren't you just using these results?

Specific Response 3:

When training and running our RF model, we used a 75/25 split (75% training and

25% testing) of our dataset to help mitigate against potential model overfitting while maintaining good model performance (low bias and RMSE). We experimented with a variety of values for the training and testing set and found the 75/25 ratio provided a balance between strong model performance, and a large test set of data to compare against. This train/test ratio also aligns with standard RF test sizes as mentioned in the SciKit-learn documentation (Pedregosa et al. 2011). After calculating our results, in order to further mitigate against potential model overfitting and to evaluate model performance on unseen data, we then went ahead and employed an additional 10-fold cross validation which resulted in an average RMSE reduction which was complimentary to our 75/25 structured model. Our 75/25 model was therefore used as the primary structure for our results since it was the original model developed and employed for bias correction, reported similar results (< 1.5 mm SWE difference) to our followup CV experiments, and was overall much more efficient to run.

Specific Comment 4:

The manuscript would be strengthened with a description of the efforts you've undertaken to mitigate temporal and spatial auto-correlation in your training and test sets. The manuscript would be strengthened with further descriptions of the efforts you've undertaken to mitigate overfitting. A comparison of training and validation errors would be an appropriate way to do this.

Specific Response 4:

In order to mitigate against spatial auto-correlation, we broke the training and testing datasets spatially as seen in Fig. 2 of the manuscript into northern and southern regions, to evaluate model performance in areas with differing magnitudes of bias and station densities. With respect to mitigating against temporal autocorrelation, we use monthly averaging of the biweekly station data which does help to some extent, however in order to fully avoid issues with auto-correlation, we would need to employ a strategy of removing stations/periods which are consistently correlated, and this would

introduce new biases in the training dataset for our model. Overall, stations are usually selected in a representative manner by the Conservation Authorities who collect measurements throughout the region, and we trust in the integrity of the station network to help mitigate this issue.

Specific Comment 5:

In Table 2, what are Year Id and Month Id? Are you using straight numerical values, cyclical temporal sin-cos pairs, 1-of-c indicators (Bishop, 1995)?

Specific Response 5:

The Year Id and Month Id predictors are 1-of-c indicators (numerical values of 0 or 1) with 0 representing the absence of either a month/year and 1 representing the presence of a month/year.

Specific Comment 6:

The water balance analysis averages melt over a watershed associated with a given stream gauge, asserting the stream gauge provides a reasonable estimate of snowmelt while at the same time neglecting evapotransportation and rainfall (actually any precipitation). Such an assertion requires that evapotransportation and subsequent precipitation are not as significant a signal as snowmelt to runoff. This may be true, but it should be backed up by analysis and references, or minimally one of these. Baseflow should also be at a minimum mentioned.

Specific Response 6:

These are fair comments and we agree that the argument can be strengthened by quantitative data. We have conducted an analysis of the dominant hydrological components across all three catchment areas, based on climate normals obtained from NRCan/CFS for the period of 1980-2010. The figure is included in this response (Fig.1 below) and could be included in supplementary material if required. It shows that in all cases average liquid precipitation (rain) during the spring freshet season exceeds potential evapotranspiration, so that it can be argued that snowmelt places a lower bound on the spring freshet volume. A nuance here is that the snowmelt peak estimated following the method of Erler et al. (2019) can (and does) exceed the streamflow peak due to routing delay within the catchment area. The peak of negative SWE differences (which is shown in Fig. 7 of the manuscript) is shown in the Figure for comparison: it is evident that the value is significantly lower than the former snowmelt estimate and does not exceed the streamflow peak. The reason is that negative SWE differences do not include water from additional snowfall during the melt period. Comparing SnoDAS SWE differences with those estimated from NRCan climate normals and streamflow, it is clear that the uncorrected SnoDAS values are unphysical, while the bias-corrected values appear reasonable. For a detailed discussion of the variables shown in the Figure and how they were processed, see section 3.2 and S2 of Erler et al. (2019); the Figure is analogous to their Fig. 2 and the datasets and methods employed are the same. The reason that this figure was not included initially is that it is based on climate normals for a period before our analysis period. Unfortunately the PET and snow depth data used in the figure are not available past 2010, so that it was not possible to update the figure. Curation of a new PET dataset (for just this figure) would be beyond the scope of this study.

Specific Comment 7:

You conclude that MBS and SLR exhibit an inability to capture year-to-year variability present in the bias, but interannual correlations are not present in the analysis. The ability of bias-correction methods particularly of the non-linear flavor to capture changes over time is arguably one of their greatest strengths, as simple offsets are more easily calculated, as you have done. A simple correlation calculation may serve as further evidence of the utility of the nonlinear method.

Specific Response 7:

We thank the reviewer for this comment and agree that the inclusion of interannual
correlations between in situ the bias corrected SWE datasets would further highlight the utility of nonlinear techniques. These results have been included in section 3.3 of the manuscript.

Specific Comment 8:

Fig 5 is hard to read with the scales and lines used, especially the in situ values, which are key to the plot. No description of shading used is given in the figure caption. Suggest changing line thicknesses/colors and/or adjusting scales, orientation, or paneling to make better use of available space.

Specific Response 8:

The Fig. 5 caption has been updated to include a description of the shaded regions (95% sampling confidence intervals). We have also updated line thickness for the in situ data to improve visibility for the reader.

References:

Bishop, C. M. (1995). Neural Networks for Pattern Recognition, Oxford University Press. Snauffer AM, Hsieh WW, Cannon, AJ, Schnorbus, MA, 2018. Improving gridded snow water equivalent products in British Columbia, Canada: multi-source data fusion by neural network models. The Cryosphere, 12(3), 891-905.

Erler, A. R., Frey, S. K., Khader, O., d'Orgeville, M., Park, Y.-J., Hwang, H.-T., Lapen, D. R., Peltier, W. R., & Sudicky, E. A. (2019). Simulating Climate Change Impacts on Surface Water Resources Within a Lake-Affected Region Using Regional Climate Projections. Water Resources Research, 55(1), 130–155. https://doi.org/10.1029/2018WR024381

Lary, D. J., Remer, L. A., MacNeill, D., Roscoe, B., & Paradise, S. (2009). Machine Learning and Bias Correction of MODIS Aerosol Optical Depth. IEEE Geoscience and Remote Sensing Letters, 6(4), 694–698. https://doi.org/10.1109/LGRS.2009.2023605

[Figure]

Lv, Z., Pomeroy, J. W., & Fang, X. (2019). Evaluation of SNODAS Snow Water Equivalent in Western Canada and Assimilation Into a Cold Region Hydrological Model. Water Resources Research, 55(12), 11166–11187. https://doi.org/10.1029/2019WR025333

Pedregosa, F., Varoquaux, G., Gramfort, A., Michel, V., Thirion, B., Grisel, O., Blondel, M., Prettenhofer, P., Weiss, R., Dubourg, V., Vanderplas, J., Passos, A., Cournapeau, D., Brucher, M., Perrot, M., and Duchesnay, E.: Scikit-learn: Machine Learning in Python, Journalof Machine Learning Research, 12, 2825–2830, 2011.

Reichstein, M., Camps-Valls, G., Stevens, B., Jung, M., Denzler, J., Carvalhais, N., & Prabhat. (2019). Deep learning and process understanding for data-driven Earth system science. Nature, 566(7743), 195–204. https://doi.org/10.1038/s41586-019-0912-1

Shen, C. (2018). A Transdisciplinary Review of Deep Learning Research and Its Relevance for Water Resources Scientists. Water Resources Research, 54(11), 8558–8593. https://doi.org/10.1029/2018WR022643

Wrzesien, M. L., Durand, M. T., Pavelsky, T. M., Howat, I. M., Margulis, S. A., & Huning, L. S. (2017). Comparison of methods to estimate snow water equivalent at the mountainrange scale: A case study of the California Sierra Nevada. Journal of Hydrometeorology, 18(4), 1101-1119.

Xue, Y., Forman, B. A., & Reichle, R. H. (2018). Estimating snow mass in North America through assimilation of AMSR-E brightness temperature observations using the Catchment land surface model and support vector machines. Water Resources Research, 54(9), p.6488.

Zahmatkesh, Z., Tapsoba, D., Leach, J., & Coulibaly, P. (2019). Evaluation and bias correction of SNODAS snow water equivalent (SWE) for streamflow simulation in eastern Canadian basins. Hydrological Sciences Journal, 64(13), 1541–1555. https://doi.org/10.1080/02626667.2019.1660780

[Figure]

**Climatology over Catchments (NRCan & WSC, 1981-2010)**

**Pic River Near Marathon (02BB003)**

Legend: Rain, Snow, PET, Snowmelt, ΔSWE, Streamflow

**South Branch Muskoka River At Baysville (02EB008)**

**Thames River At Thamesville (02GE003)**

Water Flux [mm/day]

Seasonal Cycle [month]

**Fig. 1.** Catchment water flux climatology (1981-2010) for NRCan data and stream gauge data from the Water Survey of Canada.

---

## Author Response (AR2)

**Application of machine learning techniques for regional bias correction of SWE estimates in Ontario, Canada**

**Final Comments Response**

Fraser King, Andre E. Erler, Steven K. Frey, Christopher G. Fletcher - Aug 2020

Please note the location of changes (line numbers) refer to the marked-up manuscript diff.

**General Comments**

**1) Most of the reviewer's points have been addressed but I still see need for some minor improvements. More generally, the manuscript needs a conclusion. This can be brief, but it needs to be clear where this conclusionn can be found (heading) and the conclusion needs to clarify what the contribution of this study is (beyond summarizing results and discussion).**

We agree with this suggestion and a conclusion has now been added under a new section (5) in the manuscript.

**1) Regarding comments of Rev 1**

**Rev 1 point 3) & Rev 3 point 6) These generally major criticisms should be discussed more detailed in the discussion section - was that added? If not, it needs to be added.**

We agree that these criticisms warrant a response in the paper and we have therefore included additional details as a new paragraph in the discussion section (4.2) of the manuscript on lines 28-34 of page 12 and lines 1-5 of page 13.

**2) Regarding Comments of Rev 2**

**2) The representativeness of the observation location is mentioned. A discussion of this needs to find its way into the manuscript. Are there documents and studies on that representativeness? I know form**

other snow regions, that for example, such observations are mostly taken in flat terrain and locations that may be not so wind exposed hence biased to that....etc. Such issues may help explain part of the bias.

This has now been clarified with documentation in the manuscript in section 2.1 on lines 19-22 of page 3.

**6) How does the manuscript now ensure that others don't misunderstand this as well? The methods section needs to be clear on the procedure followed.**

We have now specified this in the methods section (2.3) on lines 3-4 of page 6 to make our application of the SLR clearer to the reader.

**Rev2 point 7) and Rev 3 point 3) Training/Validation partitioning needs to be clearly explained in the methods section and not 'hidden' in the results. Please change this to make the manuscript fully transparent.**

This section has now been moved and adapted for the methods section in 2.3 on lines 24-30 of page 6.

**3) Regarding Comments of Rev 3**

**4) At least this needs to be discussed as caveat/weakness. The representativeness of the station has nothing to do with the issue of autocorrelation influencing the statistical methods. These are two different issues.**

We have now included a discussion on this topic along with references in the discussion section (4.2) on lines 12-15 on page 13 of the manuscript.

**5) Where is this clarified in the manuscript?**

Table 2 has been updated to now include this detail on the predictor types used in the DT/RF bias correction.

[revised manuscript text omitted]